# Immunomodulatory Effects of Bendamustine in Hematopoietic Cell Transplantation

**DOI:** 10.3390/cancers13071702

**Published:** 2021-04-03

**Authors:** Jessica Stokes, Megan S. Molina, Emely A. Hoffman, Richard J. Simpson, Emmanuel Katsanis

**Affiliations:** 1Department of Pediatrics, University of Arizona, Tucson, AZ 85721, USA; stokesj@email.arizona.edu (J.S.); meganm4@email.arizona.edu (M.S.M.); hoffmane@arizona.edu (E.A.H.); rjsimpson@arizona.edu (R.J.S.); 2Department of Immunobiology, University of Arizona, Tucson, AZ 85721, USA; 3Department of Nutritional Sciences, University of Arizona, Tucson, AZ 85721, USA; 4The University of Arizona Cancer Center, Tucson, AZ 85721, USA; 5Department of Medicine, University of Arizona, Tucson, AZ 85721, USA; 6Department of Pathology, University of Arizona, Tucson, AZ 85721, USA

**Keywords:** bendamustine, hematopoietic cell transplantation

## Abstract

**Simple Summary:**

Bendamustine is a chemotherapeutic agent used to treat a variety of cancers. It has recently been used in the context of allogeneic hematopoietic cell transplantation (HCT), a treatment mostly used to treat blood cancers. Given before or after transplantation of donor blood or bone marrow cells, bendamustine has been shown to reduce the side effects of the transplant, including graft-versus-host disease, where the donated cells attack the recipient’s tissues, while also promoting the anti-cancer effects of the transplant. These are exciting findings and show that bendamustine may be used to influence the immune system, called immunomodulation, in a beneficial manner. We report our research and review the available literature outlining these immunomodulatory effects of bendamustine, in hopes that it will promote further investigations utilizing this agent in allogeneic transplants, ultimately improving patient outcomes.

**Abstract:**

Bendamustine (BEN) is a unique alkylating agent with efficacy against a broad range of hematological malignancies, although investigations have only recently started to delve into its immunomodulatory effects. These immunomodulatory properties of BEN in the context of hematopoietic cell transplantation (HCT) are reviewed here. Pre- and post-transplant use of BEN in multiple murine models have consistently resulted in reduced GvHD and enhanced GvL, with significant changes to key immunological cell populations, including T-cells, myeloid derived suppressor cells (MDSCs), and dendritic cells (DCs). Further, in vitro studies find that BEN enhances the suppressive function of MDSCs, skews DCs toward cDC1s, enhances Flt3 expression on DCs, increases B-cell production of IL-10, inhibits STAT3 activation, and suppresses proliferation of T- and B-cells. Overall, BEN has a broad range of immunomodulatory effects that, as they are further elucidated, may be exploited to improve clinical outcomes. As such, clinical trials are currently underway investigating new potential applications of BEN in the setting of allogeneic HCT.

## 1. Introduction

Bendamustine (BEN) was developed in East Germany in 1963, where investigators quickly discovered its utility in treating a number of leukemias and lymphomas [1]. BEN was synthesized from nitrogen mustard and, therefore, contains a 2-chloroethylamine alkylating group and is classified as an alkylating agent. As such, BEN is able to crosslink DNA, forming repairable intra-strand lesions, as well as non-repairable inter-strand lesions. The accumulation of cross-linked DNA lesions caused by BEN administration ultimately leads to cell death, with the most robust accumulation occurring in highly proliferative cells, such as hematopoietic cells and cancerous cells. This feature of BEN is common to many chemotherapeutic compounds, including cyclophosphamide and melphalan, both of which are also alkylating agents. However, bendamustine (4-(5-[bis(2-chloroethyl)amino]-1-methyl-2-benzimidazolyl) butyric acid hydrochloride) has a unique molecular structure, consisting of a benzimidazole ring connected to a nitrogen mustard moiety, and is a purine analog in addition to a cytotoxic alkylating agent [2,3]. The benzimidazole ring is meant to confer anti-metabolite properties and is a unique feature that is not found on any other cytotoxic compound. Although few studies have investigated BEN as an anti-metabolite, it is thought that the benzimidazole ring may be able to bind to enzymes and confer biological effects in this manner [4].

While it is not clearly understood how, BEN has been reported to induce a higher frequency of and more stable double-strand breaks in DNA compared to other alkylating agents, including melphalan, cyclophosphamide, and carmustine [5,6]. Further, when comparing the effects of BEN on solid tumor cell lines (HeLa cells) and lymphoma cell lines (U2932), BEN was found to induce cell cycle arrest at lower concentrations in hematological malignancies, consistent with its clinical applications in the treatment of leukemias, lymphomas, and myelomas [7]. Among the few mechanistic studies performed on BEN, it has been shown that BEN regulates apoptotic and DNA repair pathways, inhibits mitotic checkpoints, and induces p53-dependent stress pathways [2]. BEN has similarly been shown to induce cell cycle disruption, causing G2 arrest via Cdc25 and p53 [6]. More recent studies have corroborated these findings, demonstrating that BEN induces S or G2 cell cycle arrest in a dose-dependent manner [7].

BEN is given intravenously and has a very high (95%) protein binding capacity, primarily binding to albumin, while the remaining unbound portion (5%) is biologically active. BEN undergoes significant first-pass metabolism in the liver, primarily via a cytochrome p450 enzyme complex [4], and is metabolized into two major metabolites: dihydroxy-bendamustine (HP2) and bendamustine ethyl ester (BM1EE) [8]. Nearly half of BEN is eliminated in urine in an unmetabolized form via renal excretion. BEN has a biphasic half-life, with T1/2 alpha = ~10 min and T1/2 beta = ~30 min in humans, although this pharmacokinetic property varies based on dose and species. Due to its high degree of metabolism in the liver, in addition to its renal excretion, BEN may be contraindicated in patients with renal dysfunction and severe liver damage [4,9,10]. The adverse effects of BEN are very similar to those of typical alkylating agents in the nitrogen mustard family and include nausea, vomiting, diarrhea, loss of appetite, weight loss, mucositis, immunosuppression, anemia, and thrombocytopenia [11].

Until 1990, BEN was only available in East Germany where it was marketed as Ribomustin [1]. In 2008, the United States Food and Drug Administration (FDA) approved BEN for the treatment of chronic lymphocytic leukemia (CLL) under the tradename Treanda [1]. BEN was approved for the treatment of rituximab-resistant non-Hodgkin’s lymphoma soon after [1,12]. In 2015, BEN HCl was approved in the United States under the tradename Bendeka [2]. Clinical trials are underway in the United States to assess a range of cancers that may be successfully treated with this agent [2]. Currently, BEN is indicated for CLL, multiple myeloma, breast cancer, small-cell lung cancer, and non-Hodgkin’s lymphoma [2]. Although we have limited knowledge of the precise mechanisms of action of BEN, its utility in the clinical setting has quickly expanded. It has been used as an effective lymphodepleting agent prior to infusion of chimeric antigen receptor (CAR) T-cells [13]. BEN has also been employed as a component of conditioning regimens for autologous hematopoietic cell transplantation (HCT) to treat multiple myeloma, Hodgkin’s lymphoma, mantle cell lymphoma, and other non-Hodgkin’s lymphomas [10,14,15,16,17]. More recently, BEN has been implemented in the context of clinical allogeneic HCT and has been studied in murine HCT models, both highlighting its immunomodulatory effects and suggesting favorable immunomodulation resulting in graft-versus-host disease (GvHD) and tumor control.

## 2. BEN as Pre-Transplant Conditioning in Murine Models

Bone marrow transplantation as a clinical treatment evolved from radiation studies in mice and guinea pigs in the 1950s, which demonstrated that bone marrow cell infusion can rescue animals from lethal radiation [18]. In 1956, Ford et al. confirmed that donor bone marrow cells engraft in the recipient [19]. In the same year, Barnes et al. reported that donor bone marrow cells can exert anti-leukemic effects [20]. Subsequently, in 1957, the first report of a human bone marrow transplant was published by Thomas et al. [21]. Since its inception, bone marrow transplantation or hematopoietic cell transplantation (HCT) has become a widely implemented approach to treat hematological disorders and malignancies. Experimental animal models of bone marrow transplantation have significantly furthered the field and continue to provide valuable insights into both graft-versus-host disease (GvHD) and graft-versus-leukemia (GvL) effects, leading to many of the current clinical treatments [22,23,24]. For example, steroids remain the first line therapy for management of acute GvHD [25,26,27], arising from murine studies conducted in the 1990s [28,29].

Most murine models of bone marrow transplantation rely on total body irradiation (TBI) as pre-transplant conditioning [22,24,30]. However, most clinically used pre-transplant conditioning regimens consist of chemotherapy only or a combination of chemotherapy and radiation. This has been a frequently noted limitation and weakness of murine models of HCT [22,24,30]. This is important as both the intensity and the components of the conditioning regimen have been shown to play significant roles in engraftment, immune reconstitution, non-relapse mortality, incidence and severity of GvHD, and GvL [31,32,33,34,35,36]. Pre-transplant conditioning imparts damage on host tissues, particularly epithelial tissues, resulting in the release of inflammatory signals that lead to the activation of host antigen-presenting cells, laying the groundwork for the development of GvHD [37]. Thus, it is important to explore the differential effects of conditioning regimens on this process.

There are few investigators who have reported on the addition of chemotherapy to conditioning regimens in murine models. In 1991, Hill et al. published their findings using a typical murine TBI conditioning dose in combination with cyclophosphamide (CY) [38]. They observed that, although a TNFα blockade increased survival from GvHD in a TBI only conditioning model, when CY was incorporated into the regimen, the TNFα blockade no longer reduced mortality. This highlights the importance of conditioning consideration in murine HCT models. Other groups have published on the use of busulfan with CY, simulating the conditioning regimens used clinically for myeloid malignancies [39,40]. Nevertheless, there is a paucity of published research using chemotherapy-containing preparative regimens in murine models compared to the vast body of literature that exists utilizing TBI alone.

A CY-TBI conditioning regimen has traditionally been used as conditioning for acute lymphoblastic leukemia (ALL) for adult and pediatric patients in both related and unrelated donor HCT settings, though this regimen has been associated with tissue toxicity and GvHD [31,41]. Other agents have been used clinically in combination with TBI, such as cytabarine [42], etoposide [43], and fludarabine [44]. Chemotherapeutic agents used in combination with TBI have not been previously compared to CY-TBI in experimental BMT models. To our knowledge, our group was the first to report on the substitution of cyclophosphamide with BEN in conjunction with a non-myeloablative dose of TBI as pre-transplant conditioning in both MHC-mismatched and haploidentical murine bone marrow transplant models [32].

Stokes et al. reported that administration of a comparable dose of BEN in place of CY, when used in combination with TBI as pre-transplant conditioning, was associated with significantly reduced GvHD, as evidenced by prolonged survival and decreased morbidity [32]. We confirmed these results using various numbers of total spleen cells or purified T-cells and a range of BEN and CY doses in our MHC-mismatched model. Moreover, these findings were verified in an F1→F1 haploidentical model (B6AF1→CB6F1). Additionally, in this report, we documented no difference in engraftment kinetics and confirmed in a syngeneic model that lethality was not due to conditioning regimen toxicity. In a follow-up study, our group reiterated that BEN-TBI reduced GvHD and improved survival compared to CY-TBI and further demonstrated that BEN-TBI preserved GvL effects, resulting in increased leukemia-free survival compared to CY-TBI [45]. We demonstrated BEN-TBI results in overall better outcomes than CY-TBI based on improved graft-versus-host disease-free relapse-free survival (GRFS). These studies indicate that BEN modulates the immune system to a state that suppresses GvHD while promoting GvL, through the multiple cellular mechanisms discussed below, providing compelling support for clinical study.

## 3. Clinical Application of BEN as Pre-Transplant Conditioning

BEN was first given in patients in the context of HCT as a component of a pre-transplant conditioning in the setting of autologous stem cell transplantation (ASCT). High dose chemotherapy followed by ASCT is considered an effective treatment option for a variety of malignancies, including relapsed and refractory lymphomas and multiple myeloma [46,47,48,49,50,51,52]. Due to the efficacy of BEN when utilized as chemotherapy against non-Hodgkin’s lymphoma (NHL), Visani et al., from Pesaro, Italy, published the first clinical study in which BEN was employed in a pre-transplant conditioning regimen [16]. Citing concerns of high incidence of pulmonary complications with carmustine-containing regimens [53,54,55,56], they sought to modify the widely utilized BEAM conditioning regimen, comprised of carmustine (BCNU), etoposide, cytarabine (Ara-C), and melphalan, by replacing carmustine with BEN (BeEAM). They observed complete engraftment in all patients receiving ASCT for resistant or refractory NHL or Hodgkin’s lymphoma (HL) with no transplant-related mortality (TRM) at 100 days and a complete response (CR) rate of 81%, leading them to conclude that this was a safe and effective treatment regimen that warranted further investigation [16]. In a 2014 letter to the editor, Visani et al. updated their results on these 43 patients with a median follow-up time of 41 months, reporting a 72% three-year progression-free survival (PFS) [57]. This study sparked enthusiasm for this new conditioning regimen and laid the groundwork for an increasing body of literature incorporating BEN in conditioning regimens for ASCT.

Following the demonstrated safety and effectiveness of BEN in BeEAM, Khouri et al., from MD Anderson in Houston, Texas, initiated a Phase I/II trial evaluating the use of escalating doses of BEN with historically used fixed doses of fludarabine and rituximab (BFR) as reduced intensity conditioning for allogeneic hematopoietic cell transplantation (allo-HCT) to treat patients with relapsed or resistant CLL or NHL [36]. Donors were HLA-compatible sibling donors or HLA-A, -B, -C, and -DRB1 matched unrelated donors. With 10 patients enrolled in their Phase I trial, and no observed dose-limiting toxicities, Phase II was initiated using the highest dose of BEN tested, 130 mg/m^2^ for three days. They found this regimen to be a safe and effective option, achieving a 90% two-year overall survival (OS) and 75% two-year PFS. Furthermore, they observed only 1.8% TRM at 100 days, reduced myelosuppression, and only 11% Grade II–IV acute graft-versus-host disease (aGvHD). They also observed 26% two-year extensive chronic GvHD (cGvHD). Twenty-three percent of patients recovered counts without growth factor support and 88% of patients did not require platelet transfusions. Patients recovered an absolute neutrophil count of >0.5 × 10^9^/L at a median of zero days after allo-HCT. The authors noted that this reduced myelosuppression was striking in comparison to their historical data [36]. In 2017, the same group updated their results of this study, now including 69 patients with a median follow-up time of five years. They reported a 74% five-year OS and a 60% five-year PFS. They confirmed their previously published results with continued low GvHD rates and reduced myelosuppression [58] (Table 1). In 2017, Khouri et al. reported long-term follow-up of 26 CLL patients receiving BFR prior to allo-HCT compared to 63 patients receiving fludarabine, cyclophosphamide, and rituximab (FCR) conditioning, demonstrating significant improvements in three-year OS (82% vs. 51%) and three-year PFS (63% vs. 27%), as well as significantly reduced incidence of severe neutropenia (62% vs. 97%). They also observed reduced TRM and reduced incidence of Grade III/IV aGvHD [35] (Table 1). Although thus far only MD Anderson has reported on the incorporation of BEN in conditioning regimens for allogeneic HCT, these results are notable and warrant further studies. They also recently initiated a trial that focuses on PT-BEN but will include patients who receive BEN in their pre-transplant conditioning regimen (Table 2). To our knowledge, there are no published clinical reports combining BEN with total body irradiation in an allogeneic HCT setting, although the MD Anderson PT-BEN trial (NCT04022239) will employ BEN + TBI conditioning with fludarabine. These clinical results using BFR corroborate our published murine studies using BEN + TBI, indicating BEN acts on the immune system in a manner that promotes GvL and suppresses GvHD, while resulting in reduced myelosuppression.

## 4. Post-Transplant BEN in Murine Models

While allo-HCT is an effective treatment modality for patients with hematological malignancies and disorders, finding an HLA-matched donor remains an impediment to implementation of this treatment for many patients. This is especially challenging for minority patients [59,60,61]. As such, more and more institutions have begun implementing haploidentical (haplo) HCT [62,63,64,65,66,67,68,69,70]. This both increases donor availability and decreases delays involved in finding and collecting stem cells from an unrelated donor. The use of post-transplant CY (PT-CY) has revolutionized the use of haploidentical HCT, significantly reducing GvHD [71,72,73,74,75,76,77]. Although several studies have focused on graft manipulation to reduce GvHD, including CD3ε+ or TCR-αβ+ depletions [64,78,79,80], PT-CY is favored by many as it can be employed easily at any center. However, concerns remain regarding high relapse rates, especially with nonmyeloablative conditioning regimens, and viral reactivations reported with PT-CY [81]. In an attempt to address these concerns, we published the first murine study utilizing post-transplant BEN (PT-BEN) in place of PT-CY following haploidentical HCT [33]. Mice were conditioned with reduced intensity or myeloablative TBI followed by haplo-HCT. PT-CY significantly reduced GvHD in our haploidentical models, resembling what is seen clinically. Administration of PT-BEN in lieu of PT-CY, utilizing doses comparable in terms of maximum tolerated dose, did not impair engraftment, provided equivalent protection from GvHD, and promoted a superior GvL effect against a B-cell leukemia. Additionally, PT-BEN was less myelosuppressive than PT-CY, with significantly higher white blood cell counts through Day +12 and higher myeloid to lymphoid ratios through Day +13. Importantly, PT-BEN treated mice, in contrast to PT-CY treated mice, never became neutropenic following transplant and we observed higher hemoglobin, platelet, and red blood cell counts up to Day +14 with myeloablative conditioning [33]. This again promotes the notion that BEN has significant immunomodulatory effects that may be beneficially harnessed in the context of HCT.

## 5. Clinical Application of BEN Post-Transplant

Based on our preclinical findings, we hypothesized that PT-BEN can safely replace PT-CY in human haplo-HCT, with potential advantages including earlier engraftment and enhanced immune reconstitution leading to improved GRFS. Our clinical trial at the University of Arizona, which was posted on ClinicalTrials.gov in December 2016 (NCT02996773), was the first human trial applying post-transplant BEN. The objective of this ongoing Phase I trial is to evaluate the safety of substituting PT-BEN for PT-CY following haploidentical HCT. PT-CY is typically administered on Days +3 and +4 following HCT. Our Phase I is a standard 3 + 3 dose escalation design with six dose level cohorts. De-escalation of CY and concomitant escalation of BEN on Day +4 was deemed the safest approach with which to initiate the trial. The first three cohorts consisted of a combination of sequentially reduced doses of CY and increased doses of BEN, to a maximal dose of 90 mg/m^2^, on Day +4 post-HCT with the dose of CY on Day +3 remaining unchanged. The subsequent cohorts, currently enrolling, involve Day +3 progressive substitution of BEN for CY, resulting in complete substitution on both days in Cohort 6, with patients receiving 90 mg/m^2^ each day. Some of the patients in our early cohorts of the trial were included in two broad haplo-HCT publications from our institution [63,82] and the interim analysis of our Phase I trial was recently published, including the first three cohorts of our study [83]. In this interim analysis of the first three cohorts, Katsanis et al. demonstrated that patients achieved trilineage engraftment earlier as PT-BEN escalated, consistent with other murine and clinical data indicating reduced myelosuppression with BEN. The median time to an absolute neutrophil count of 1.0 × 10^9^/L was achieved earlier in each progressive PT-BEN cohort. Similarly, the later PT-BEN cohorts demonstrated earlier platelet engraftment and required fewer platelet and red blood cell transfusions. All PT-BEN patients showed complete donor chimerism. No Grade III/IV aGvHD or cGVHD was seen in patients receiving PT-BEN. We saw no dose limiting toxicity or non-relapse mortality. There was no difference in the incidence of bacteremia between patients receiving PT-BEN compared to PT-CY controls and no patients developed fungal infections. CMV reactivation was significantly reduced in the three cohorts receiving PT-BEN (12%) versus comparable patients receiving PT-CY (71%). With a median follow-up of greater than 25 months, the overall survival at two years was 83.3% and graft-versus-host disease-free relapse-free survival was 71.1% [83] (Table 2). We are also investigating immune reconstitution differences between PT-CY and PT-BEN in our trial, although these data are not yet mature enough to draw conclusions from and are currently unpublished. Although a small study thus far, these results are encouraging and suggest PT-BEN warrants further study.

Recently, other centers have started to initiate trials utilizing PT-BEN. Moiseev et al. from St. Petersburg, Russia published two abstracts on a dose de-escalation study of PT-BEN. The study (NCT02799147) intended to enroll three cohorts of 10 patients each, receiving 140, 100, or 70 mg/m^2^ BEN on Days +3 and +4 in a de-escalation study [84,85]. The first cohort receiving 140 mg/m^2^ per day was closed after six patients due to severe infectious complications. Enrollment in this study is now complete at 26 patients, 5 patients with ALL and 21 with acute myeloblastic leukemia (AML). They reported that 73% of patients experienced cytokine release syndrome (CRS), contributing to their 43% non-relapse mortality. They additionally observed severe chronic GvHD in 70% of patients, although this was better controlled when other immunosuppressive agents were given in addition to the PT-BEN, with 40% of those patients experiencing chronic GvHD. They observed modest prevention of aGvHD, with Grade III/IV aGVHD observed in 43%, 30%, and 33% of patients by respective cohort. Similar to our study, patients received myeloablative conditioning, although a different regimen. However, while all of the transplants in our study were haploidentical, in this study, only 27% received a haploidentical, 58% a matched unrelated, and 15% a matched sibling donor. They found that 92% of patients engrafted with 62% achieving a measurable residual disease (MRD) negative remission, resulting in 29%, 40%, and 70% one-year OS, respectively (Table 2). Of note, none of the ALL patients achieved long-lasting complete remission. While these results may appear less encouraging than our data, it is important to note that Moiseev’s trial enrolled only patients with at least 5% blasts in their bone marrow prior to HCT, while in our trial most patients were in complete remission prior to BMT. Their more effective dose of 100 mg/m^2^ is similar to 90 mg/m^2^ per day that we are currently studying. These data indicate that careful consideration should be given to the dose of BEN administered [83,84].

More recently, a group from MD Anderson in Houston also initiated a trial (NCT04022239) utilizing PT-BEN (Table 2). Their dose-escalation of PT-BEN begins by progressively replacing Day +3 PT-CY, with all patients receiving PT-BEN on Day +4, similar to Cohorts 4–6 in our Phase I trial [83]. The trial description on ClinicalTrials.gov does not indicate the doses to be used in this study, but it does state the study will include only adult patients 18–65 years old with hematological malignancies using mismatched or haploidentical donors. This differs from our trial, which also includes pediatric patients (8–45 years). The conditioning regimens also differ, and, interestingly, the Khouri et al. trial description indicates that some patients will also receive BEN as part of their conditioning regimen. This group has previously published the outcomes of allogeneic transplants conditioned with BEN, fludarabine, and rituximab [35,36,58]. They indicate that, in this trial, a subset of patients, depending on diagnosis, will receive BEN, fludarabine, and TBI, with or without rituximab, depending on CD20 status. As this study matures, it will provide valuable information on the combination of BEN conditioning and post-transplant administration.

We summarize the ongoing post-transplant bendamustine clinical trials in Table 2. To our knowledge, this is a complete account of the published and ongoing studies using PT-BEN in humans. As the two ongoing trials mature, they will provide valuable information regarding the safety and efficacy of BEN as a post-transplant treatment, as well as how it may compare to the standard of PT-CY.

## 6. Immunomodulatory Effects of BEN

While investigating the effects of BEN on GvHD, GvL, survival, and other clinical outcomes is crucial to determining patient care, it is also important to evaluate the effects of BEN on specific immune cells in an attempt to understand its specific immunomodulatory effects.

### 6.1. Myeloid Derived Suppressor Cells (MDSCs)

Myeloid derived suppressor cells (MDSCs) are a heterogenous population of myeloid cells with a suppressive function, defined phenotypically in mice as CD11b+Gr-1+ cells. These cells have been shown to be pivotal in murine GvHD, as adoptive transfer of MDSCs generated in vitro or isolated from in vivo models have been shown to reduce GvHD in an allogeneic bone marrow transplantation setting [86,87,88,89,90]. Additionally, in humans, greater frequency of MDSCs in the donor graft has been associated with reduced aGvHD [91,92,93]. As such, chemotherapeutic agents that have an immunomodulatory effect on MDSCs may be of great interest in efforts to control GvHD.

In our preclinical investigations on the use of BEN pre-transplant, we demonstrated increased myeloid cell infiltration in the intestines post-transplant [32]. We additionally showed a higher frequency of MDSCs in the bone marrow, spleen, and blood following BEN-TBI conditioning. Moreover, when MDSCs were depleted using an anti-Gr-1 antibody, the survival difference between BEN-TBI and CY-TBI was no longer significant. Conversely, administration of granulocyte colony stimulating factor (GCSF), which expands MDSCs, widened the survival difference between BEN-TBI and CY-TBI. We concluded the attenuation of GvHD achieved with BEN-TBI conditioning was, at least in part, attributable to its effects on MDSCs [32]. Similarly, we reported that PT-BEN increases the frequency of MDSCs in the blood following haploidentical BMT [33], indicating that given before or after transplant, BEN results in increased proportions of MDSCs.

Additionally, we demonstrated that in vitro generation of murine BM-derived MDSCs in the presence of BEN significantly increases their suppressive function against activated T-cells [33], although no studies have shown increased suppressive function of MDSCs isolated from BEN-treated mice. These in vitro data, along with our in vivo murine data showing increased MDSC numbers in tissues and the clinical data showing rapid and sustained neutrophil engraftment when BEN is utilized in the allogeneic HCT setting, indicate that this agent may have a distinct effect on the myeloid compartment.

### 6.2. Effector T-cells and T Regulatory Cells

It has been shown that the phenotype of T-cells post-allogeneic bone marrow transplantation can have a crucial effect on GvHD, as well as on GvL. Th1 versus Th2 skewing, co-stimulatory and co-inhibitory molecule expression, and other T-cell phenotypic factors have been implicated as major players in the pathogenesis of GvHD [94,95,96,97]. Additionally, T regulatory cells (Tregs) have been shown to significantly reduce GvHD [98,99,100]. As such, when evaluating BEN as an immunomodulatory agent in HCT, it is important to consider its effects on T-cells.

In an allogeneic BMT model utilizing PT-BEN, we reported that particularly in a myeloablative setting, PT-CY compared to PT-BEN results in increased absolute numbers of CD4+ T-cells, with no differences in absolute numbers of CD8+ T-cells [33]. We also recently reported on the role and fate of donor T-cells following BEN-TBI conditioning in a major histocompatibility complex mismatched murine transplant model. We demonstrated that BEN-TBI does not result in considerable differences in T-cell phenotype following transplant when compared with CY-TBI conditioning. Nevertheless, we found that T-cells harvested after transplantation from surviving BEN-TBI conditioned mice were tolerant to host, but not third-party, MHC antigens. Lastly, we determined that the enhanced GvL effect seen with BEN-TBI conditioning is dependent on T-cells, indicating that, although these T-cells are tolerant to host MHC, they can still exert anti-leukemic effects [45]. We posited T-cell tolerance may be due in part to the effect of the increased number of MDSCs, discussed above, as MDSCs can induce T-cell tolerance [101,102]. We also found decreased donor T-cell infiltration in the intestines post-transplant [32]. In our earlier studies, we demonstrated that T-cells stimulated with CD3/CD28 activating beads were significantly less proliferative when cultured with BEN [33]. This is consistent with the in vivo observations, indicating that BEN results in less proliferative T-cells and thus, reduced presence of T-cells in GvHD target organs. Therefore, BEN may have direct effects on effector T-cells in addition to indirect effects though other cell populations.

Although Treg play a significant role in the suppression of GvHD [103,104,105], we have not found specific effects of BEN on this cell subset. We investigated in vitro generation of T regulatory cells in the presence of BEN and did not observe a skewing in phenotype or suppressive function [45]. Additionally, we demonstrated that donor Treg are not required for the reduction of GvHD seen with BEN-TBI or PT-BEN [32,33]. This is in contrast to what others have shown in murine models of PT-CY [106]. In our PT-BEN model, we also showed that PT-CY resulted in greater numbers of Treg in the blood following transplant than PT-BEN [33].

### 6.3. B-Cells

B-cells have been implicated in the development of chronic graft-versus-host disease [107,108,109]. However, B regulatory cell (Breg) frequency has been shown to be a predictor of reduced GvHD and adoptive transfer of Bregs can mitigate GvHD [110,111]. Bregs are thought to reduce GvHD partially through secretion of IL-10 [111]. Interestingly, in 2016, Lu et al. demonstrated using Ramos cells, a human B cell line derived from a Burkitt lymphoma, that BEN inhibited proliferation of the cell line and its IgM secretion, but concurrently increased its production and secretion of IL-10 [112]. This group further showed that when peripheral blood mononuclear cells from healthy human donors were cultured with BEN, B-cell production of IL-10 was increased. Using inhibitors to explore the mechanism of this enhanced IL-10 production, the researchers found that BEN has this effect through the p38 MAP kinase-Sp1 pathway [112]. We also found that murine B-cells were significantly less proliferative when activated with LPS and cultured in the presence of BEN [33]. While we found no difference between PT-BEN and PT-CY in terms of absolute numbers of B-cells in the blood at various time points following transplant [33], these data indicate that BEN skews the function of B-cells in a manner that may have significant anti-inflammatory effects.

### 6.4. Dendritic Cells

Host dendritic cells (DCs) persist long enough following HCT to stimulate naïve donor T-cells and are, therefore, critical in the pathogenesis of GvHD, particularly the initiation phase [113,114]. In recent studies, we reported BEN-TBI paradoxically results in significantly greater absolute numbers of host DCs, yet reduced GvHD compared to CY-TBI. We also report significant changes in the composition of host DCs compared to CY-TBI during the peri-transplant period [115]. Flow cytometric analysis of splenic DCs found that the proportion of plasmacytoid DCs, as well as CD8α+ Type 1 conventional DCs (cDC1s), CD103+ cDC1s, and pre-cDC1s were significantly increased in BEN-TBI compared to CY-TBI. This study offered additional insight into potential mechanistic pathways for the reduction of GvHD seen with BEN-TBI conditioning. Most prominently, pre-cDC1s were ~5-fold greater in number in mice conditioned with BEN-TBI compared to CY-TBI. cDC1s are implicated in ameliorating GvHD through clonal deletion of self-reactive T-cells and play an important role in promoting anti-cancer cytotoxic CD8+ T-cell responses [116,117,118]. In this same study, we used Batf3 knockout mice as recipients, demonstrating that Batf3-dependent host DCs (CD8α+ and CD103+ cDC1s) are not necessary for reduced GvHD following BEN-TBI conditioning [115]. Interestingly, pre-cDC1s were similarly found to be ~5-fold greater in number in this transgenic model and were inversely associated with GvHD severity in Batf3 knockout mice conditioned with BEN-TBI. Although we hypothesize BEN may be exerting its beneficial effects partially through pre-cDC1s, there are no studies to date investigating this DC precursor in the context of GvHD and GvL, so its role in GvHD protection remains to be elucidated. We also demonstrated an increase in Flt3 receptor tyrosine kinase expression on host DCs conditioned with BEN-TBI compared to CY-TBI, suggesting that this upregulation of Flt3 receptor may contribute to the favoring of cDC1 development compared to other DC subsets [115].

In a follow-up study on the effect of BEN on DCs, our group further demonstrated that murine bone marrow-derived dendritic cells (BMDCs) generated following brief exposure to BEN exhibited a concentration-dependent increase in pre-cDC1 frequency and Flt3 receptor tyrosine kinase surface expression. In line with these findings, BEN has previously been shown to modulate cytokine secretion in B-cells via the p38 MAP kinase pathway [112], which is activated downstream of Flt3 [119]. Further, Flt3 activation can suppress autophagy [120], which promotes long-term cross-presentation in murine DCs [121], and increase DC lifespan [122]. This is suggestive of a potential mechanism by which BEN induces increased expression of Flt3 and pathways by which enhanced Flt3 activation may alter DC phenotype and function in the context of alloreactivity. We further characterized these BMDCs observing that BEN exposure induces a regulatory phenotype, with lower iCOS-L expression, higher PD-L1 expression, and significantly reduced secretion of the pro-inflammatory cytokines IL-6, TNFα, CCL5, and CCL2. However, BEN exposure does not similarly inhibit the secretion of the anti-inflammatory cytokine IL-10. Furthermore, generation of human monocytic-DCs following brief exposure to BEN similarly developed a concentration-dependent increase in Flt3 receptor expression and an accompanying decrease in phospho-STAT3. Finally, we demonstrated BMDCs generated following exposure to a high concentration of BEN result in robust alloreactive T-cell proliferation followed by programmed cell death of ~50% of all alloreactive T-cells in culture (submitted). These data indicate that BEN has a significant immunomodulatory effect on dendritic cell proportions, phenotype, and function, potentially contributing to its protective effects in the setting of HCT.

### 6.5. Immunomodulatory Pathways

It is also important to consider, aside from cell type-specific effects, how BEN may more globally affect immunologically relevant pathways. Interestingly, Iwamoto et al. studied the biochemical interactions of BEN with signal transducer and activator of transcription (STAT) proteins [8]. STAT proteins function downstream of receptor tyrosine kinases and are critical regulators of pathways of inflammation, proliferation, differentiation, apoptosis, survival, and immune responses [123]. One member of this family of proteins, STAT3, is known to be constitutively activated in many types of hematopoietic and solid tumors [124]. Canonical activation of STAT3 requires phosphorylation at the Tyrosine residue 705 (Y705), after which STAT3 is able to dimerize at the pY705 residue and the SH2 domain. The pSTAT3 dimer is then able to translocate into the nucleus and regulate gene expression. Iwamoto et al. found the chloride groups of BEN directly bind to the SH2 domain of STAT3 and inhibit the interaction between the phosphorylated tyrosine 705 residue and the SH2 domain, effectively inhibiting canonical STAT3 activation [8]. Furthermore, they showed that BEN inhibits this interaction due to its binding affinity for cysteine residues, as mutations at Cys550 and Cys712 resulted in decreased sensitivity to BEN [8]. Supportive of this immunomodulatory effect, we observed that when human monocytes are exposed to BEN for 4 hours prior to DC generation, this brief exposure results in significantly reduced pY705-STAT3 expression by the resulting dendritic cells at the end of culture. This indicates that BEN stably binds to and inhibits canonical STAT3 signaling (submitted). As STAT3 is involved in many different processes, the many potential immunological consequences of this inhibition remain to be seen.

## 7. Conclusions

BEN is a versatile drug, showing promise as chemotherapy for a variety of cancers, as a conditioning regimen component for autologous HCT, and as a lymphodepleting agent. A body of literature has only recently started to accumulate regarding the immunomodulatory properties of BEN, as summarized in Figure 1. We have studied the effects of BEN when employed both pre- and post-BMT in multiple murine models. We have consistently observed decreased GvHD, increased GvL, and significant changes to the proportion and phenotype of multiple immune cell types. Additionally, in vitro studies have shown BEN can increase the suppressive function of MDSCs, skew DC generation toward cDC1s, increase DC Flt3 expression, increase B-cell production of IL-10, inhibit STAT3 phosphorylation, and suppress B- and T-cell proliferation. Clinically, BEN is being utilized in patients receiving HCT as conditioning and as a post-transplant treatment to reduce GvHD. These studies have shown promising results and, as the trials mature, will provide further insight into the effects BEN has on the immune system. It is already clear BEN has a large range of immunologic effects that, as we better understand them, may be exploited for better patient outcomes.

## Figures and Tables

**Figure 1 cancers-13-01702-f001:**
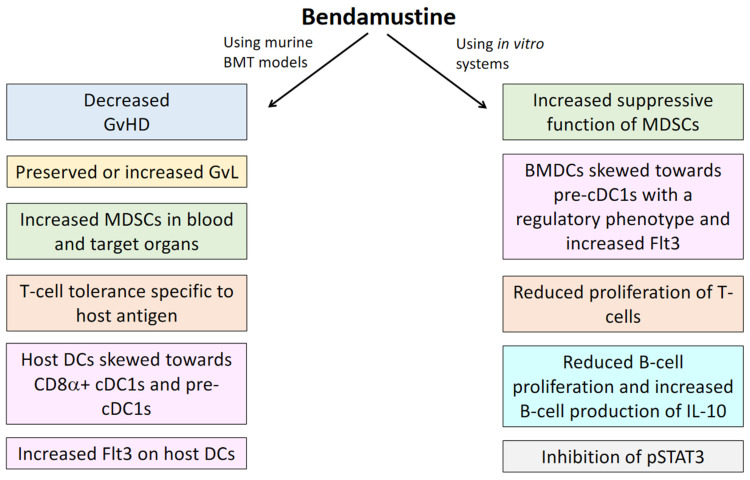
Summary of the immunomodulatory effects of bendamustine observed in murine models and in vitro systems.

**Table 1 cancers-13-01702-t001:** Clinical trials using pre-transplant bendamustine in allogeneic HCT.

	N	Age	Donor Graft	Disease	Remission Status %	Regimen	Engraft %	aGvHD II-IV %	cGvHD %	NRM %	OS %	PFS %
**Khouri** (Houston, Texas) **2009**- NCT00880815 Phase I/II Dose escalation of BEN (70, 90, 110, and 130 mg/m^2^)	69 closed	30-72	MSD or MUD PBSC or BM	CLL Lymph	42 CR 46 PR 12 RD	RIC FLU-BEN- -Ritux	100	17	31	9	74 @ 5y	60 @ 5yr
**Khouri** (Houston, Texas) **2009**- NCT00880815; NCT00899431 Evaluation of BFR conditioning compared to FCR	26 closed	49-72	MSD or MUD PBSC or BM	CLL	8 CR 54 PR 38 RD	RIC FLU-BEN- -Ritux or FLU-CY-Ritux	100	23	45	8	82 @ 3y	63 @ 3y

BEN = bendamustine, MSD = matched sibling donor, MUD = matched unrelated donor, PBSC = peripheral blood stem cells, BM = bone marrow, CLL = chronic lymphocytic leukemia, CR = complete remission, PR = partial remission, RD = refractory disease; RIC = reduced intensity conditioning, FLU = fludarabine, Ritux = rituximab, Engraft = engraftment; aGvHD = acute graft versus host disease, cGvHD = chronic graft versus host disease, NRM = non-relapse mortality, OS = overall survival, PFS = progression free survival; BFR = bendamustine fludarabine rituximab; FCR = fludarabine cyclophosphamide rituximab; CY = cyclophosphamide.

**Table 2 cancers-13-01702-t002:** Clinical trials using post-transplant bendamustine in allogeneic HCT.

	N	Age	Donor Graft	Disease	Remission Status %	Regimen	Engraft %	aGvHD III-IV %	cGvHD %	NRM %	Relapse %	OS %	EFS %
**Katsanis** (Tucson, Arizona) **2016**- NCT02996773 Phase I/Ib Dose-escalation of PT-BEN day +4 (20-60-90 mg/m^2^)/ de-escalation of PT-CY Day +3 CY	9 ongoing	9–42	Haplo BM	Leuk Lymph	33 CR1 22 CR2 22 >CR2 22 PR	MAC TBI-FLU or BU-FLU-MEL	100	0	0	0	29 @ 2yr	83 @ 2y	71 @ 2yr
**Moiseev** (St. Petersburg, Russia) **2016**- NCT02799147 Phase I/II De-escalation of PT-BEN days +3, +4 (140-100-70mg/m^2^)	26 closed	20–56	MSD or MUD or Haplo PBSC	Leuk	RD	MAC BU-FLU	92	43 30 33	70	43	19	29 40 70 @ 1y	29 40 27 @ 1y
**Khouri** (Houston, Texas) **2019**- NCT04022239 Phase I/II Day +4 BEN Dose-escalation of PT-BEN day +3/de-escalation of PT-CY	ongoing	18–65	Haplo or MMUD PBSC	Leuk Lymph	?	RIC FLU-MEL-TBI or FLU-BEN-TBI ± Ritux	-	-	-	-	-	-	-

BEN = bendamustine, CY = cyclophosphamide; Haplo = haploidentical, BM = bone marrow, MSD = matched sibling donor, MUD = matched unrelated donor, PBSC = peripheral blood stem cells, MMUD = mismatched unrelated donor, Leuk = leukemia, Lymph = lymphoma, CR = complete remission, PR = partial remission, RD = refractory disease; MAC = myeloablative conditioning, TBI = total body irradiation, FLU = fludarabine, BU = busulfan, MEL = melphalan, RIC = reduced intensity conditioning, Ritux = rituximab, Engraft = engraftment; aGvHD = acute graft versus host disease, cGvHD = chronic graft versus host disease, NRM = non-relapse mortality, OS = overall survival, EFS = event free survival; ? = unknown.

## Data Availability

No new data were created or analyzed in this study. All data is available in the cited articles. Any additional data for our studies is available by request of the corresponding author.

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
