# Peer review of "Immunomodulatory Effects of Bendamustine in Hematopoietic Cell Transplantation"

_cancers, 2021, doi:10.3390/cancers13071702_

Round 1
Reviewer 1 Report
In this review, the authors summarized the immunomodulatory properties of bendamustine in the context of hematopoietic cell transplantation and consequent therapeutic implications. They reported how pre- and post-transplant use of bendamustine in murine models resulted in reduced GvHD and enhanced GvL with particular attention to the effects of immunomodulation on different cell populations including T cells, myeloid derived suppressor cells (MDSCs) and dendritic cells. In addition they reported in vitro studies in which Bendamustine enhances the suppressive function of MDCs, skews DCs toward cDC1s, enhances Flt3 expression on DCs, increases B-cell production of IL-10, inhibits STAT3 activation, and suppresses proliferation of T- and B-cells.
The manuscript is clear and easy to understand. The organization of the paragraphs is acceptable. The experimental data on the possible immunomodulatory mechanisms induced by Bendamustine are very interesting. I would recommend summarizing them, if possible, in a figure that would be very helpful to the reader.
Reviewer 2 Report
This article is a comprehensive review of the immunomodulatory effects of bendamustine in hematopoietic cell transplantation, and I convince that it absolutely contributes to the field of transplantation.
However, since the topic of this paper is complex and often difficult to understand from the text alone, it would have been better and reader-friendly if there were figures showing the comprehensive immune network and the impact of bendamustine on it, especially about GVHD and GVL effects.
In the introduction, the authors said "BEN has also become a frequently employed component of conditioning regimens for autologous hematopoietic stem cell transplantation, particularly for multiple myeloma", but I do not agree with the authors about this. The use of bendamustine in ASCT of MM is not common at this time around the world.
Table 1 included only the clinical trials using post-transplant bendamustine, but a table including trials of other conditioning regimens using bendamustine may be more informative.
The characters in table 1 and table 2 are too small.
6. Immunomodulatory effects of BEN
The paragraph about Treg, "Though Treg play a significant role...." should not be placed in the section "6.1 Myeloid-derived suppressor cells (MDSC)". The authors should create another section and describe BEN effect on T cells in another section.
Reviewer 3 Report
Authors described about review of bendamustine. Some concerns remains thus please corrected the manuscript.
- Bendamustine is used for pre-transplant conditioning regimen in patients with mantle cell lymphoma. Please, add this point and cited the paper (Bone marrow transplantation. 2020;55:1076-1084).
- Authors described the differences between PTCY and PT bendamustine. The more detailed biological differences (e.g. immune cell reconstitution )should be added.
- Please, add the features of MDSC fraction (e.g. Gr1+, CD11b+).
- MDSC is associated with regulatory T cell. Regulatory T cell is major components of immune suppressor. Authors should be added this points.
Round 2
Reviewer 3 Report
The manuscript is corrected appropriately.